## [Peer Review File · Development (Cambridge, England)]

Reciprocal inhibition of Wnt signaling pathways pattern the interconnection of epithelial tubules in the regenerating zebrafish kidney

Caramai N. Kamei, William G.B. Sampson, Carolin Albertz, Oliver Aries, Amber Wolf, Rohan M. Upadhyay, Samuel M. Hughes, Heiko Schenk, Frederic Bonnet, Bruce B.W. Draper, Kyle W. McCracken, Denise K. Marciano, Leif Oxburgh and Iain A. Drummond
DOI: 10.1242/dev.205074

Editor: James M Wells

Review timeline

Original submission:	3 July 2025
Editorial decision:	7 September 2025
First revision received:	22 December 2025
Accepted:	17 January 2026

Original submission

First decision letter

MS ID#: dev.205074

MS TITLE: Mutually repressive Wnt signaling pathways pattern the interconnection of epithelial tubules in the regenerating zebrafish kidney

AUTHORS: Caramai N. Kamei; William G.B. Sampson; Carolin Albertz; Oliver Aries; Amber Wolf; Rohan Upadhyay; Samuel Hughes; Heiko Schenk; Frederic Bonnet; Bruce B.W. Draper; Kyle McCracken; Denise Marciano; Leif Oxburgh; Iain Alexander Drummond

Dear Dr Drummond,

I have now received all the referees' reports on the above manuscript, and have reached a decision. The referees' comments are appended below, or you can access them online: please go to: View Reviewer Comments

As you will see, the referees express considerable interest in your work, but have some significant criticisms and recommend a substantial revision of your manuscript before we can consider publication. If you are able to revise the manuscript along the lines suggested, which may involve further experiments, I will be happy to receive a revised version of the manuscript. Your revised paper will be re-reviewed by one or more of the original referees, and acceptance of your manuscript will depend on your addressing satisfactorily the reviewers' major concerns. Please also note that Development will normally permit only one round of major revision. If it would be helpful, you are welcome to contact us to discuss your revision in greater detail. Please send us a point-by-point response indicating your plans for addressing the referees' comments, and we will look over this and provide further guidance.

Please attend to all of the reviewers' comments and ensure that you upload both a 'clean' version of your Word file, along with a highlighted version clearly showing where you have made changes in the revised manuscript. Please avoid using 'Tracked changes' in Word files as these are lost in PDF conversion. I should be grateful if you would also provide a point-by-point response detailing how

you have dealt with the points raised by the reviewers in the 'Response to Reviewers' box. If you do not agree with any of their criticisms or suggestions please explain clearly why this is so.

Reviewer 1

SUMMARY OF THE ADVANCE MADE IN THIS PAPER AND ITS POTENTIAL SIGNIFICANCE TO THE FIELD

The manuscript by Kamei et al outlines the role of both canonical and non-canonical wnt signaling in the regenerating kidney tubules. The study outlines many impactful and novel findings that are relevant to our general understanding of how epithelial sheets fuse, including cell shape changes involving actin dynamics and constriction and how basal membrane protrusions drive fusion events. The study is also considered to be highly mechanistic and outlines antagonistic signaling roles for the canonical and non-canonical wnt pathway, making it relevant to a broad readership. The data presented appears to be highly rigorous and supports the major conclusions of the study in most cases. Prior to publication, I ask the authors to address the following minor points with existing data.

SUGGESTIONS TO AUTHORS

Minor points

In all figures and legends, please indicate what arrows and arrow heads are pointing to. (Figure 1, Figure 4 arrowheads point to different morphologies, etc.)

While it is assumed that all images are representative, knowing the number of examples where similar morphologies were observed would strengthen the conclusions. Quantification is nicely carried out for curvatures and protrusions but not for in situ data.

Figure 2 and 3 *lhx1a-egfp* is not reduced in the IWR1 treatment, however, dramatically reduced in *wnt4* and *wnt9b* mutants by ISH. Please confirm that the staining of *lhx1a* in Figure 3 A-C is ISH and the localization is expected. Please clarify the difference in detection and the reason for this.

Figure 3 y-axis in G and H are difficult to read. It is also unclear if the numbers are from *lhx1a-egfp* or ISH. Please clarify.

Figure 3 new nephron generation is thought to be reduced, but Figure 3 A-C shows the tissue is there and the *lhx1a* is reduced. This is not described carefully in the results.

Similarly in Figure 3 D-F show a reduction in EdU staining that is not well elaborated or quantified.

The discussion could be strengthened by adding conclusion sentences to some of the paragraphs that discuss the presented data in the paper in the context of what is discussed and cited in the literature.

Reviewer 2

SUMMARY OF THE ADVANCE MADE IN THIS PAPER AND ITS POTENTIAL SIGNIFICANCE TO THE FIELD

This manuscript examines how tubular interconnections form during zebrafish kidney regeneration. The authors propose that canonical and non-canonical Wnts, acting via paracrine and autocrine mechanisms, trigger (or act in parallel with) Src and Rac to regulate protrusive behaviors, cell shape, orientation, proliferation, and polarity, processes that may or may not directly influence interconnection formation.

While the work addresses an important biological process, the presentation is difficult to follow due to the complexity of the data, unclear descriptions of analyses, and some inconsistencies between figures and text. I recommend simplifying the narrative, providing clearer explanations of experimental design and interpretation, improving figure quality, and expanding the model figure. Ideally, the model will have multiple panels including wildtype, *Wnt4*, *Wnt9b* and *Fzd9* mutants and gene expression/activity/proliferation in each scenario. If some of the confusion here is based on different stages of development, this should also be included in the model. Specific concerns are outlined below.

SUGGESTIONS TO AUTHORS

Major Concerns

1. Spatial and Temporal Context of Imaging

Determining image orientation and spatial context is challenging throughout the manuscript. For example: In some cases in wildtype tissue, *Lhx1gfp*⁺ expression is polarized and only found in cells closest to the distal tubule that are largely Edu negative (Figures 1, 2 and 4). In others, *Lhx1Gfp* is everywhere including Edu-positive cells (Figure 3). So is *Lhx1GFP* expression spatially located or is it not? How are we to spatially orient ourselves. In wildtype tissues, one can use proliferation but in some mutants, proliferation shifts spatially, so that doesn't work. One could use the distal tubule for orientation, yet in several figures (e.g., 6b and 6d) it is not visible (In 6b, a line is drawn, but the cells are not visible and in 6d, there is no distal tubule indication or visible).

This is problematic because much of the analysis depends on knowing the precise region shown. For instance, proliferation appears expanded toward the distal tubule in *Fzd9* mutants in Figure 6, but not in Figure 5, where the distal tubule is clearly visible. The discrepancy between these figures needs explanation.

2. *Wnt4*, *Fzd9* and Proliferation

The authors claim *Fzd9* blocks proliferation, yet *Fzd9* appears to be expressed in proliferating cells. This seems contradictory. I think their model is that expansion of *Wnt4* expression and canonical *Wnt* activity in mutants leads to expanded proliferation. But the expanded proliferation is observed in the domain of normal *Wnt4* activity, not the expanded domain. The precise relationship of canonical *Wnt* activity to *Wnt4* and *Fzd9* expression is unclear. Either canonical activity is normally present in the proliferating cells and promotes proliferation (which makes the most sense based on the loss of proliferation in the *IWR1* treatments) or it is present in *Wnt4* expressing cells where proliferation does not normally occur (which is what their model claims). But if the latter is the case, then why does expanded *Wnt4* promote proliferation? Clarification is needed to reconcile this observation. Ideally any clarification would include expanding and clarifying the model figure.

3. Expression Patterns in Relation to Previous Work

In Kamei et al. (2019), *Fzd9* was reported to be expressed throughout the newly formed nephron, including in *Wnt4*-expressing cells. In the current study, the authors state the two genes have non-overlapping expression. Is this a temporal difference, or has the understanding changed based on new methods or conditions?

4. Autocrine vs. Paracrine *Wnt4* Signaling

There is an inconsistency between the text (line 277: autocrine) and the Figure 8 legend (paracrine) regarding *Wnt4* signaling. Which is correct-or is it both? For example: If *Wnt4* signals to new nephron cells that form protrusions and do not proliferate or express *Fzd9b*, in my understanding that would be autocrine. But if *Wnt4* signals to *Fzd9b*-expressing cells (possibly those with high *Tcf/Lef* reporter activity) to promote proliferation, that would be paracrine.

5. Nature of Newly Formed Nephrons in *Wnt* Mutants

The significant reduction in new nephron formation in *Wnt9* and *Wnt4* mutants raises questions about the structures being analyzed. There seem to be several possibilities including: 1) these are indeed new nephrons but they are formed via a *Wnt*-independent pathway. 2) *Wnts* regulate proliferation of a progenitor population required for all new nephrons. In this scenario, new nephrons form but reduced proliferation limits their absolute numbers. Or 3) The *Lhx1*⁺ structures are a pre-existing, perhaps senescent cell type that existed prior to the injury. This distinction is essential, as claims about canonical *Wnt* activity's role in invasive behavior depend on whether the analyzed structures are truly newly formed nephrons and, if they are, how "normal" they are.

6. It is not clear to me whether any of the observed defects lead to issues with new nephron fusion with distal tubule. The data provided is very unclear, the statements in the results are vague and there is no quantification.

Minor Concerns

1. The legend for Figure 1c is missing or incomplete.
2. In Figure 3, there is substantial variability in new nephron formation in controls; in one cohort, the average is close to *Wnt9b* mutant levels.
3. In Figure 4D', the image does not appear to be a high-magnification view of the protrusion shown in panel D.

First revision

Author response to reviewers' comments

We thank you and our reviewers for the time and effort spent on our manuscript. We feel the critique and responses have strengthened our conclusions and made the paper a bit easier to understand. It is a complicated story but one of significant interest, we feel, to the organogenesis community. We have addressed each of the points raised with new data or revision of the text. The changes we have made are stated below and highlighted in the revised text in red.

Responses to review:

Reviewer 1: SUMMARY OF THE ADVANCE MADE IN THIS PAPER AND ITS POTENTIAL SIGNIFICANCE TO THE FIELD

The manuscript by Kamei et al outlines the role of both canonical and non-canonical wnt signaling in the regenerating kidney tubules. The study outlines many impactful and novel findings that are relevant to our general understanding of how epithelial sheets fuse.

We thank the reviewer for their constructive comments.

Minor points

In all figures and legends, please indicate what arrows and arrow heads are pointing to. (Figure 1, Figure 4 arrowheads point to different morphologies, etc.)

Thank you for bringing this to our attention. All figures have been reviewed and simplified for this revision, using a minimum of annotation for specific features. Annotation in figures 1 and 4 have been made consistent.

While it is assumed that all images are representative, knowing the number of examples where similar morphologies were observed would strengthen the conclusions. Quantification is nicely carried out for curvatures and protrusions but not for in situ data.

We have quantified the overexpression of *wnt4* and *lef1* in the *fzd9b* mutants and generated two new graphs of the data in supplemental figure 4. The quantification approach is presented in the Methods. In brief, we quantified the area in sections of gene expression in wildtype and *fzd9b* mutant new nephrons using ImageJ regions of interest and present the data as expression area in square microns.

Figure 2 and 3 *lhx1a-eGFP* is not reduced in the IWR1 treatment, however, dramatically reduced in *wnt4* and *wnt9b* mutants by ISH. Please confirm that the staining of *lhx1a* in Figure 3 A-C is ISH and the localization is expected. Please clarify the difference in detection and the reason for this.

In figure 2, fish are treated with IWR1 for 24 hours and sacrificed for analysis. In figure 3, fish have never expressed *wnt4* or *wnt9b* (germline homozygous mutants). We do not expect a dramatic reduction of the *lhx1a:eGFP* reporter in the short term IWR1 experiment (GFP protein half-life would result in persistent expression) while the germline wnt mutants would have stronger phenotypes. Also, we do not claim that *lhx1a:eGFP* is specifically a wnt reporter although it may be affected by wnt signaling.

The staining for *lhx1a* in Figure 3 A-C is wholemount ISH. This figure, and graph below in E and F, represents the reduction in new nephron differentiation after injury in *wnt4* and *wnt9b* mutants, presumably due to the reduction in Wnt signaling. What may be surprising is that any nephrons are induced at all in the mutants. As elaborated below, this is likely due to low-level expression of *wnt4* and *wnt9b* paralogs (*wnt4b* and *wnt9a*).

We have shown previously, and in new data added to this manuscript, that *wnt9a* is weakly expressed after injury (Kamei et al., 2019) and not affected by *wnt9b* mutation in the adult zebrafish kidney (this ms., supplemental figure 3). Based on the survival of *wnt4* and *wnt9b* homozygous mutants to adulthood and absence of any significant difference in nephron number between mutants and wildtype (supplemental figure 2A,B), we argue that these genes are not required for mesonephric kidney development and that they represent regeneration-inducible *wnt* paralogs.

Figure 3 y-axis in G and H are difficult to read. It is also unclear if the numbers are from *lhx1a-egfp* or ISH. Please clarify.

We have replaced the graphs with larger lettering for the Y-axis. These graphs represent *lhx1a+* (ISH) new nephrons induced after gentamicin injury. We added clarifying text in the figure legend and the results text.

Figure 3 new nephron generation is thought to be reduced, but Figure 3 A-C shows the tissue is there and the *lhx1a* is reduced. This is not described carefully in the results.

Zebrafish mesonephric kidney development is not affected by mutation in *wnt4* or *wnt9b*; this is shown by counting nephrons (with glomeruli staining for nephrin) in adult homozygous mutants (supplemental figure 2). From this we conclude that the initial development of the adult kidney is not impaired in *wnt4* and *wnt9b* homozygous mutants. This is mentioned in the results section with reference to supplemental figure 2. So, kidney tissue is indeed there in the *wnt4* and *wnt9b* mutants. Also, in addition to nephrons, much of the kidney is composed of immune cells which is the "bone marrow" of the adult fish, and this tissue is not noticeably affected by *wnt* mutations.

We have added new text to the results to state this more clearly.

Similarly, in Figure 3 D-F show a reduction in EdU staining that is not well elaborated or quantified.

The main purpose of this experiment was to assess formation of invasive basal protrusions and whether Wnt signaling was required for mesenchymal invasiveness. We wanted to know whether genetic mutant phenotypes were consistent with IWR1 induced loss of invasive basal protrusions. We find that mutant phenotypes are consistent with *wnt* inhibitor treatment and add the number of observations of the loss of basal protrusions in mutants to the text and figure legend.

Cell proliferation in mutants was not so clear cut. Our prior publication demonstrated a *wnt* requirement for proliferative outgrowth of new nephrons (Kamei et al., 2019) by showing a complete block in nephron cell proliferation and tubule outgrowth by prolonged treatment with IWR1. In the data here on individual mutants, we see variable loss of proliferation in *wnt4* mutant aggregates, presumably due to continued expression of *wnt9b*. EdU incorporation is reduced in *wnt4* mutants but not eliminated. This data is shown more fully in slices of the confocal stack, comparing wt (supplemental movie 4) with *wnt4* mutant (Supplemental movie 5) and referenced in the text. For *wnt9b* mutant aggregates, EdU incorporation was not performed as part of the experiment. Given the size of the aggregates, it is likely that some proliferation does occur as in *wnt4* mutants and may be due to *wnt9a* paralog expression (supplemental figure 3).

Given the complexity of having at least four (and possibly more) *wnt* paralogs expressed in the new nephrons, the best conclusion may be drawn from IWR1 treatment (complete block of proliferation with 4-day treatment (Kamei et al., 2019)). We state this in the revised text.

The discussion could be strengthened by adding conclusion sentences to some of the paragraphs that discuss the presented data in the paper in the context of what is discussed and cited in the literature.

Thank you for suggesting this; we have reviewed the discussion added more

context for the results.

Reviewer 2: SUMMARY OF THE ADVANCE MADE IN THIS PAPER AND ITS POTENTIAL SIGNIFICANCE TO THE FIELD

This manuscript examines how tubular interconnections form during zebrafish kidney regeneration. The authors propose that canonical and non-canonical Wnts, acting via paracrine and autocrine mechanisms, trigger (or act in parallel with) Src and Rac to regulate protrusive behaviors, cell shape, orientation, proliferation, and polarity, processes that may or may not directly influence interconnection formation.

While the work addresses an important biological process, the presentation is difficult to follow due to the complexity of the data, unclear descriptions of analyses, and some inconsistencies between figures and text. I recommend simplifying the narrative, providing clearer explanations of experimental design and interpretation, improving figure quality, and expanding the model figure.

We have extensively revised the text for clarity and added new data to strengthen our conclusions. Inconsistencies have been addressed and resolved to the best of our ability.

Ideally, the model will have multiple panels including wildtype, Wnt4, Wnt9b and Fzd9 mutants and gene expression/activity/proliferation in each scenario. If some of the confusion here is based on different stages of development, this should also be included in the model. Specific concerns are outlined below.

We remade the summary figure 8 according to the reviewer's suggestion and incorporate different panels for wildtype, *wnt4*, and *fzd9* mutants which showed the clearest and strongest phenotypes. The model also now includes a consideration of temporal stages of interconnection that better rationalize the data we present.

Major Concerns

1. Spatial and Temporal Context of Imaging

Determining image orientation and spatial context is challenging throughout the manuscript. For example: In some cases, in wildtype tissue, Lhx1gfp⁺ expression is polarized and only found in cells closest to the distal tubule that are largely Edu negative (Figures 1, 2 and 4). In others, Lhx1Gfp is everywhere including Edu-positive cells (Figure 3).

Tissue orientation is critical in evaluating the data in the paper. We include a new figure 1A with a diagram of the tissue interaction in new nephron formation to help orient readers.

There are also several guides in the data to orient newly forming nephrons. New nephrons are uniquely marked by expression of *lhx1a:eGFP* in the figures. Expression was typically strongest at the junction of new and old nephron tubules. In all figures, new nephron fusion to distal tubule is oriented toward the bottom of the panel. We have shown previously that all new nephrons form exclusively on distal tubules (Kamei et al., 2019) and abut, as a blind ended tube, prior to fusion. New nephron basal invading surfaces also were the only location of invasive protrusions (central or laterally displaced as in *fzd9b* mutants). Presence of the distal tubule also provides spatial context to orient the new nephrons. The reviewer is correct in pointing out that in figure 3, *lhx1a:eGFP* expression extends further away from the point of interconnection but here, the distal tubule is clearly marked, and basal invasive protrusions are clearly visible. Several animated 3D MIPs of wildtype and mutant new nephrons are part of the submission (movies 1, 3, 10, 11, and 12).

We agree that orienting the images is especially critical for evaluating the proliferation change in *fzd9b* mutants. As suggested by the reviewer, we have changed figure 6B to

show an example of the distal tubule / new nephron interaction, and where the lateral displacement of invasive protrusions in the *fzd9b* mutant is also obvious.

The most conclusive data on ectopic proliferation in the *fzd9b* mutant is in quantification of EdU+ nuclei in the bottom 6 μm of multiple wt and *fzd9b* new nephrons. These images were 3D cropped to isolate the bottom 6 μm volume of the invasive interface (data used to generate figure 6E). To best illustrate the difference in the *fzd9b* mutant, we include four new movies (supplemental movies 6-9) with this revision that show the new nephron-distal tubule interface volume of the new nephron and the ectopic proliferation in *fzd9b* mutants. All image stacks used for this analysis will be publicly available on the MDIBL Omero image server.

This is problematic because much of the analysis depends on knowing the precise region shown. For instance, proliferation appears expanded toward the distal tubule in Fzd9 mutants in Figure 6, but not in Figure 5, where the distal tubule is clearly visible. The discrepancy between these figures needs explanation.

As noted above, single slices of nephrons do not tell a complete story. Figure 5F represents a single, submicron slice of a new nephron in a *fzd9b* *-/-*; *tcflf:dGFP* mutant transgenic. This image demonstrates the expansion of the canonical wnt reporter *tcflf:dGFP* expression in the *fzd9b* mutant homozygotes. The data on proliferation is supported by analysis of 14 full 3D stacks of *fzd9b* mutant new nephrons, analyzed as in figure 6E.

2. Wnt4, Fzd9 and Proliferation

The authors claim Fzd9 blocks proliferation, yet Fzd9 appears to be expressed in proliferating cells. This seems contradictory. I think their model is that expansion of Wnt4 expression and canonical Wnt activity in mutants leads to expanded proliferation. But the expanded proliferation is observed in the domain of normal Wnt4 activity, not the expanded domain. The precise relationship of canonical Wnt activity to Wnt4 and Fzd9 expression is unclear. Either canonical activity is normally present in the proliferating cells and promotes proliferation (which makes the most sense based on the loss of proliferation in the IWR1 treatments) or it is present in Wnt4 expressing cells where proliferation does not normally occur (which is what their model claims). But if the latter is the case, then why does expanded Wnt4 promote proliferation? Clarification is needed to reconcile this observation. Ideally any clarification would include expanding and clarifying the model figure.

Our model is that expression of *fzd9b* in early new nephron aggregates establishes signals that block/antagonize canonical wnt-dependent cell proliferation in cells closest to the source of canonical Wnt ligand, the distal tubule. In the absence of *fzd9b*, proliferation occurs ectopically in distal tubule-adjacent cells of new nephrons. The reviewer makes a good point that what is missing from our original models is a proposed temporal series of events that can account for all the data.

We re-examined the expression of *fzd9b* during nephron formation and present new data in figure 5 A, B relevant to this and the point below. In brief, *fzd9b* is expressed in undifferentiated nephron progenitor cells prior to aggregation (figure 5A). This is also supported in our single cell RNA seq data set (Tang et al 2017). In new aggregates, prior to cell polarization, *fzd9b* remains uniformly expressed in two-cell thick new nephron aggregates (figure 5B). When new nephrons polarize and elongate, *fzd9b* is repressed in cells closest to the distal tubule, the source of the wnt ligand Wnt9b (figure 5C; (Kamei et al., 2019)). We include additional examples of this *fzd9b* pattern in polarized aggregates in new supplemental data (supplemental figure 4).

Despite the fact that the new nephron domain immediately adjacent to the distal tubule should be the site of highest canonical Wnt signaling, EdU incorporation does not occur here in wildtype nephrons. We conclude there must be an inhibitor of canonical Wnt proliferation expressed in these cells. In *fzd9b* mutants, EdU incorporation is observed in this basal cell layer, and we conclude that *fzd9b* is a proliferation inhibitory signaling that is missing. This ectopic proliferation is dependent on canonical Wnt signaling since

it is blocked by IWR1 treatment of *fzd9b* mutant fish (figure 6E).

To the question:

How can Fzd9b normally block proliferation (in wt fish) if its mRNA expression is becoming repressed in the domain closest to the distal tubule, the domain in closest proximity to *wnt9b* expressing distal tubules? Based on our observations and suggestions by the reviewer to include a temporal aspect in our model, we propose that *fzd9b* expression induces a persistent non-proliferative differentiated state on basal cells that lasts longer than the persistence of *fzd9b* mRNA. This would be expected due to Frizzled9b membrane protein half-life, and persistence of *fzd9b* signaling. Membrane proteins may persist for many hours after encoding mRNA is gone and persistent signaling may be established early on in these cells by repressive *fzd9b* signal transduction.

Why do some new nephron cells go on to proliferate at a distance from the distal tubule where *fzd9b* remains expressed in polarized aggregates? We propose that the induction of the ligand, *wnt4*, in new nephrons may shift the balance toward driving cell proliferation to extend the nephron where *fzd9b* expression is waning, based on the observation that no tubules grow out from the initial *wnt4* mutant aggregates (figure 3E).

What is clear from the data is that *fzd9b* acts as a repressor of canonical Wnt-driven cell proliferation in the basal cell layer of newly patterned nephrons (figure 6). This data makes the point that *fzd9b* antagonizes not only canonical wnt target gene expression (figure 5, supplemental figure 5) but also a canonical wnt signaling-dependent process, cell proliferation (figure 6). It is also clear that *wnt4* expression is required to pattern *fzd9b* expression in new nephrons and repress *fzd9b* mRNA since *fzd9b* is strongly and uniformly expressed in multi-layered *wnt4* mutant new nephron aggregates (figure 5). We include new additional examples of this defect in new supplemental figure 4.

3. Expression Patterns in Relation to Previous Work

In Kamei et al. (2019), Fzd9 was reported to be expressed throughout the newly formed nephron, including in Wnt4-expressing cells. In the current study, the authors state the two genes have non-overlapping expression. Is this a temporal difference, or has the understanding changed based on new methods or conditions?

We agree with the reviewer that this is a temporal difference based on a closer analysis of *fzd9b* expression in pre-aggregate single cells, newly formed small aggregates, and polarizing cell aggregates (new figure 5) where a pattern of repression was previously overlooked.

4. Autocrine vs. Paracrine Wnt4 Signaling. There is an inconsistency between the text (line 277: autocrine) and the Figure 8 legend (paracrine) regarding Wnt4 signaling. Which is correct-or is it both? For example: If Wnt4 signals to new nephron cells that form protrusions and do not proliferate or express Fzd9b, in my understanding that would be autocrine. But if Wnt4 signals to Fzd9b-expressing cells (possibly those with high Tcf/Lef reporter activity) to promote proliferation, that would be paracrine.

We agree with the reviewer that the best term would be autocrine, and we have corrected the text to make this more accurate. The *wnt4* mutation induces a phenotype in the cells that express *wnt4*.

5. Nature of Newly Formed Nephrons in Wnt Mutants

The significant reduction in new nephron formation in Wnt9 and Wnt4 mutants raises questions about the structures being analyzed. There seem to be several possibilities including: 1) these are indeed new nephrons but they are formed via a Wnt-independent pathway. 2) Wnts regulate proliferation of a progenitor population required for all new nephrons. In this scenario, new nephrons form but reduced proliferation limits their absolute numbers. Or 3) The Lhx1+ structures are a pre-existing, perhaps senescent cell type that existed prior to the injury. This distinction is

essential, as claims about canonical Wnt activity's role in invasive behavior depend on whether the analyzed structures are truly newly formed nephrons and, if they are, how "normal" they are.

New nephrons that we examine after injury in *wnt4* and *wnt9b* mutants are not present prior to injury and are likely induced by (low level) expression of the paralogous genes *wnt4b* and *wnt9a*. We demonstrated previously (Kamei et al., 2019) that both *wnt9a* and *wnt9b* are induced after kidney injury and here we add new data showing that baseline *wnt9a* expression is not altered *wnt9b* mutants (supplemental figure 3). We also provide new data demonstrating a complete loss of *wnt9b* expression in *wnt9b* homozygotes. Our efforts to grow adult double mutants for *wnt9a/b* and *wnt4/wnt4b* failed due to embryonic lethality.

The rare new nephron aggregates that do form in mutants are definitely not normal; they exhibit a mutant phenotype. *wnt4* mutants show loss of invasive protrusions, failure in tubule outgrowth / extension, and failure in cell polarization and lumen formation. In *wnt9b* mutant "escapers", new nephron aggregates appear more normal but do not make basal cell protrusions and show a much restricted expression of the *lhx1a:eGFP* transgene. The significance of this latter finding is not fully known but implies absence of signals required for this *lhx1a* promoter transgene expression in the mutant. We include the image of a *wnt9b* mutant new nephron in figure 3E to demonstrate the lack of basal protrusions and cite the number of times we observed this.

6. It is not clear to me whether any of the observed defects lead to issues with new nephron fusion with distal tubule.

We demonstrated previously that prolonged, complete inhibition of canonical wnt signaling (IWR1 treatment from day 3 to day 7) completely blocked new nephron invasion and connection (Kamei et al., 2019). We added reference to this prior work in the current text to make our point on interconnection clearer. Shorter IWR1 exposures in this work address the cellular mechanism of failed interconnection. Canonical Wnt signaling deficiencies (IWR1, *wnt4*, *wnt9b* mutants) lead to loss of the mesenchymal invasive phenotype at the new nephron - distal tubule interface (figure 3) and failed invasion of the target tubule, particularly clear in *wnt4* mutants (figure 3E) and quantified in IWR1 experiments (figure 2F).

Absence of orthogonally polarized basal protrusions in the *fzd9b* mutant leads to aberrant, laterally directed migration of fusing cells and absence of orthogonal tubule interconnections (compare 3D projections in supplemental movie 2 with supplemental movies 11 and 12 and figures 7H and I). *fzd9b* mutants and ROCK inhibitor treated fish also exhibit a failure in cell shape change that prevents or distorts new nephron connections. Failed connection in *fzd9b* mutants was confirmed by quantifying the absence of filtered dextran in new nephrons compared to wildtype and also with evidence of massively distended tubules after injury, an expected outcome of obstructed connections.

Epithelial interconnection is a complex process requiring multiple interacting signaling systems, given the cell rearrangements that must occur while maintaining tubule integrity. We acknowledge it is difficult to study and can really only be analyzed in vivo or in 3D culture. We feel that the tools, results, and functional assays we brought to bear on the question in vivo (mutants, transgenic reporters, high resolution imaging, signal blocking drugs, and kidney functional assays/dextran filtration) are an important and rigorous contribution to our understanding of the interconnection process, about which little is currently known.

Minor Concerns

1. The legend for Figure 1c is missing or incomplete.

Thank you; we have corrected this with the new figure 1.

2. In Figure 3, there is substantial variability in new nephron formation in controls; in one cohort, the average is close to Wnt9b mutant levels.

There is substantial variability in new nephron formation in both controls and mutants. Nonetheless there is a significant difference in new nephron number for wt, +/-, and -/- conditions for each wnt mutant (figure 3G,H).

3. In Figure 4D', the image does not appear to be a high-magnification view of the protrusion shown in panel D.

Yes this is correct. 4D' is a different slice from the same image stack as 4D which shows the same actin filament distribution. We now note this in figure legend.

We hope the new data and revised text in this submission address the concerns raised by reviewers. Again, we thank you for the time and effort you put into this review.

Second decision letter

MS ID#: dev.205074R1

MS TITLE: Reciprocal inhibition of Wnt signaling pathways pattern the interconnection of epithelial tubules in the regenerating zebrafish kidney

AUTHORS: Caramai N. Kamei; William G.B. Sampson; Carolin Albertz; Oliver Aries; Amber Wolf; Rohan Upadhyay; Samuel Hughes; Heiko Schenk; Frederic Bonnet; Bruce B.W. Draper; Kyle McCracken; Denise Marciano; Leif Oxburgh; Iain Alexander Drummond

ARTICLE TYPE: Research Article

Dear Dr Drummond,

I am happy to tell you that your manuscript has been accepted for publication in Development, pending our standard publication integrity checks.

Reviewer 1

SUMMARY OF THE ADVANCE MADE IN THIS PAPER AND ITS POTENTIAL SIGNIFICANCE TO THE FIELD

Beautiful paper. Nicely revised too.

Reviewer 2

SUMMARY OF THE ADVANCE MADE IN THIS PAPER AND ITS POTENTIAL SIGNIFICANCE TO THE FIELD

This is a beautiful study on the mechanisms regulating tubule fusion in the zebrafish nephron. The authors have addressed my previous concerns. I support publication at this time.